# Grazing and light modify *Silene latifolia* responses to nutrients and future climate

**Maria-Theresa Jessen**[ID][1,2,3]*, **Harald Auge**[1,2], **W. Stan Harpole**[1,3,4], **Yann Hautier**[5], **Anu Eskelinen**[1,3,6]

1 German Centre for Integrative Biodiversity Research–iDiv Halle-Jena-Leipzig, Leipzig, Germany,
2 Department of Community Ecology, Helmholtz Centre for Environmental Research—UFZ, Halle, Germany,
3 Department of Physiological Diversity, Helmholtz Centre for Environmental Research–UFZ, Leipzig, Germany, 4 Martin Luther University Halle-Wittenberg, Halle (Saale), Germany, 5 Department of Biology, Ecology and Biodiversity Group, Utrecht University, Utrecht, The Netherlands, 6 Department of Ecology and Genetics, University of Oulu, Oulu, Finland

* maria-theresa.jessen@ufz.de

**Data Availability Statement:** The data are available in Figshare under the following DOI https://doi.org/10.6084/m9.figshare.21369714.v1.

## Abstract

Altered climate, nutrient enrichment and changes in grazing patterns are important environmental and biotic changes in temperate grassland systems. Singly and in concert these factors can influence plant performance and traits, with consequences for species competitive ability, and thus for species coexistence, community composition and diversity. However, we lack experimental tests of the mechanisms, such as competition for light, driving plant performance and traits under nutrient enrichment, grazer exclusion and future climate. We used transplants of *Silene latifolia*, a widespread grassland forb in Europe, to study plant responses to interactions among climate, nutrients, grazing and light. We recorded transplant biomass, height, specific leaf area (SLA) and foliar carbon to nitrogen ratio (C:N) in full-factorial combinations of future climate treatment, fertilization, grazer exclusion and light addition via LED-lamps. Future climate and fertilization together increased transplant height but only in unlighted plots. Light addition increased SLA in ambient climate, and decreased C:N in unfertilized plots. Further, transplants had higher biomass in future climatic conditions when protected from grazers. In general, grazing had a strong negative effect on all measured variables regardless of added nutrients and light. Our results show that competition for light may lead to taller individuals and interacts with climate and nutrients to affect traits related to resource-use. Furthermore, our study suggests grazing may counteract the benefits of future climate on the biomass of species such as *Silene latifolia*. Consequently, grazers and light may be important modulators of individual plant performance and traits under nutrient enrichment and future climatic conditions.

## Introduction

Grasslands are increasingly exposed to concurrent impacts of climate change, nutrient enrichment and changes in grazing practices [1]. Singly and in concert these global change drivers can affect plant community composition and diversity [2–4]. In a changing environment,

**Funding:** This work was supported by the Flexpool program of the German Centre for Integrative Biodiversity Research (iDiv) – Halle, Jena, Leipzig [grant no. 34600565-11] and by a research grant from the Finnish Academy to A.E. [project 29719]. There was no additional external funding received for this study. https://www.idiv.de/en/research/flexpool.html https://www.aka.fi/en/ The funders had no role in study design, data collection and analysis, decision to publish, or preparation of the manuscript.

**Competing interests:** The authors have declared that no competing interests exist.

plant communities should harbor species which are able to tolerate direct and interactive pressures of these drivers [5,6]. To predict and understand overall community responses, it is therefore essential to identify how abiotic and biotic changes interactively influence plant persistence and competitive ability. One approach to assess plant vulnerability to current global change drivers is measuring intraspecific trait responses which can relate to plant performance [7,8]. However, while some studies have examined individual species' traits and performance in response to different land-use types [9,10], studies examining the combined impacts of nutrient enrichment, grazing and climate change are scarce [but see 8,11], particularly in temperate systems. In addition, not much is known about the mechanisms driving these trait-based plant responses.

Nutrient enrichment directly increases plant available nutrients and can thus increase plant growth and biomass accumulation [12,13] and lead to larger SLA and higher relative foliar N content [14,15]. Fertilization-induced enhanced plant growth and leaf enlargement can reduce light availability in the understory [16,17], disfavoring small-statured species with resource-conservative traits (i.e., small SLA, high foliar C:N) [18]. Consequently, tall-statured species with resource-acquisitive traits (i.e., large SLA, low foliar C:N) should perform better and dominate in nutrient-rich conditions [15,19–21]. In a study conducted in forest understory [22], plant height increased and SLA decreased in response to light addition, but were unaffected by fertilization, suggesting that light was more limiting than nutrients for trait manifestation [23]. However, results from other systems, such as grasslands, might differ, because they lack shade created by trees and plant species inhabiting grasslands might be less well adapted to shaded conditions [24]. However, little is known about grassland plant responses to light, and how light interacts with other factors to control plant performance and traits, as no field studies have directly added light. Many grassland systems are also intensively grazed by mammalian herbivores that can further alter resource availability and plant responses to resource availability.

Grazing can reduce fertilization-induced light limitation which has, on the community level, a beneficial effect for biodiversity [16]. On the individual plant level, the direct effect of grazing on plant growth depends on grazing intensity [25] and plant traits (i.e., stature, leaf size, leaf nutritional composition) that affect vulnerability to herbivory [11,26–28]. In general, grazing affects plant growth by decreasing height and biomass which can reduce plant performance [26,29,30], but grazing can also stimulate plant growth in a density-dependent manner [31] and eventually lead to overcompensation [32,33]. However, abiotic conditions such as soil fertility can alter grazing effect on plants [34,35] by allowing greater compensation for grazing damage in resource-rich conditions [36,37]. This emphasizes that the net effect of grazing on plant biomass and trait manifestation is likely dependent on resource availability (i.e., light and nutrients). Furthermore, the interaction of herbivory and resources on plant responses may be altered by changing climate.

Climate change can alter plant-herbivore interactions through direct effects on herbivores but also directly affecting plants [38]. Likewise, climate change may modify plant-resource interactions, for example, due to accelerated nutrient cycling in response to warmer temperatures [39]. For temperate grasslands in Central Germany, climate change forecasts predict warmer temperatures throughout the growing season, increased spring and autumn precipitation, and reduced precipitation and increasing risk of drought spells during summer [40–42]. Warmer temperatures and water surplus at the beginning of the growing season can stimulate plant growth, increasing height and biomass [43,44], and in general promoting traits of fast resource-use [45]. Herbivores, by targeting taller plants [28,46], can dampen the effect of warming on plant traits and thus alter plant performance in response to warmer temperatures [8,11]. On the other hand, reduced precipitation during summer can cause physiological stress

with impacts on plant performance [47–49]. To cope with drought induced stress, plants may profit from traits that minimize water loss such as smaller SLA [7,50–52]. Some studies have examined the interactive effects of climate change and nutrient enrichment in grassland systems [53–55], but only a few have investigated how climate change and nutrient enrichment interact with herbivory [56].

We used transplants of a widely distributed and well-studied European grassland species, *Silene latifolia* [57,58; hereafter *Silene*], to study plant responses to interactions among climate, nutrients, grazing and light. We chose *Silene* as our representative of typical forbs in temperate grasslands because, besides being common and widely distributed, it is palatable to herbivores [59,60], important for pollinators [61], and has also been shown to respond to environmental changes [62] and substantially contributes to the local recruitment from seed in our study system (Jessen & Eskelinen, personal observation). We therefore expect this species to be highly responsive to resource manipulations and herbivory, and changes in its performance should reflect to community dominance relationships. We grew *Silene* individuals under full-factorial combination of nitrogen, phosphorus and potassium (NPK) fertilization, light addition and herbivore (sheep) exclusion, and replicated these treatments under ambient and future climatic conditions. We directly manipulated light in the field, using modern LED lamps that mimic natural sun light. Our future climate of seasonally altered temperature and precipitation mimicked projected climate for Central Europe [63]. We measured two common traits (SLA and foliar C:N) linked to resource-use strategies [64–66], competitive ability [19], and palatability to herbivores [46,67]. As a measure of performance, we collected data on plant aboveground biomass and plant height. We asked the following questions: 1) Are the effects of nutrient and light additions, future climate, or their interactions on *Silene* performance greater in the absence of grazers? 2) Does increased nutrient and light availability, future climate, or their interactions affect *Silene* foliar traits? We made the following predictions: 1a) Nutrient and light additions will increase *Silene* height and biomass more in the absence of grazers, because competition for these resources is greater under dense vegetation (i.e., without grazing). 1b) This pattern will further depend on future climate which can either amplify or reduce competition for nutrients and light, depending on whether it increases or decreases the biomass of surrounding vegetation. 2a) Light addition will dampen the effect of nutrient addition on *Silene* SLA and foliar C:N since fertilization-induced light limitation is relaxed. 2b) Future climate will interact with fertilization and light addition on *Silene* SLA and foliar C:N, because it can either amplify or reduce competition for light, depending on whether it increases or decreases the biomass of surrounding vegetation.

## Material and methods

### Study site

Our experiment is embedded in the Global Change Experimental Facility (GCEF) at Bad Lauchstädt Research Station (51˚22060 N, 11˚50060 E), Germany. In this area the long-term mean annual precipitation is 489 mm and the mean annual temperature is 8.9˚C [63]. The soils are fertile Haplic Chernozems [63,68]. The GCEF combines research on different land-use types with ambient and future climate regimes in a split-plot design [63]. For this study we used ten extensively grazed pastures within the GCEF which represent species-rich grassland vegetation of dryer regions of Central Germany [63]. These pastures are grazed two to three times a year by a herd of about 20 German black-headed sheep, depending on the amount of vegetation biomass [63]. In 2018, 2019 and 2020 grazing was limited to two grazing events per year because of extreme drought conditions [69,70]. Each pasture has a size of 16 x 24 m and is grazed by the sheep with high intensity for 24 hours [63]. This short term, high intensity

grazing is a common practice to maintain open grasslands in Germany [71]. To prevent other larger, naturally occurring grazing animals such as deer from entering the experimental site, the entire area was fenced. Small mammalian herbivores such as the European hare (*Lepus europaeus*) can enter the experimental area.

Five of the pastures were randomly assigned to the ambient climate, the other five to the future climate scenario. For the climate manipulation, pastures under future climatic conditions are equipped with metal roof structures with foldable roof and side covers, as well as with a controllable irrigation system [63]. To control for any effect of the roof structure, pastures in ambient conditions have similar structures but with non-foldable roof and side covers. The intent of the climate treatment is to increase temperatures and to simulate decreasing precipitation in the summer and increasing precipitation in the spring and fall, consistent with predictions for this region [63]. The future climate treatment increases air and soil temperature using passive night warming for which the roof and side covers are automatically closed during the night. The amount of temperature increase varies over season and years; on average the daily minimum temperature increased by 1.14˚C [63]. To simulate increasing spring and autumn precipitation the weekly precipitation at ambient blocks is used as a reference to which ten percent is added to simulated the increase in precipitation in future blocks [63]. To simulate decreasing summer precipitation, future blocks receive 80 percent of the weekly precipitation that falls on ambient blocks [63].

## Experimental design

We established our experiment in 2017 using a full factorial design of herbivore (sheep) exclusion and nutrient enrichment in each of the ten pastures (ambient and future climate). In each of the ten pastures, we established four $1.4 \times 1.4$ m plots that were randomly assigned to the following treatment combinations: 1) unfertilized and grazed (e.g. control), 2) fertilized and grazed, 3) unfertilized and fenced, 4) fertilized and fenced. This resulted in 40 experimental plots, arranged in four-plot groups (called "blocks" in the statistical analyses), half of which received ambient climate and the other half future climate treatment. In later March-May and June we added a slow-release NPK fertilizer mixture (Haifa Multicote 2 M 40-0-0 (40% N), Super Triple Phosphate TPS (45% $P_2O_5$), potassium sulfate fertilizer (50% $K_2O$, 45% $SO_3$)) to each plot assigned to fertilization treatment. This corresponds to an addition of 10 g N, 10 g P and 10 g K m$^{-2}$ [see the protocol of 72] per growing season to experimentally relax nutrient limitation. The fertilization treatment started in May 2017, while the herbivore exclusion treatment started three months later, at the end of August. We used rectangular, mobile metal fences of $1.8 \times 1.8$ m size, 82 cm height and 10 cm mesh size to exclude the sheep from plots assigned to the herbivore exclusion treatments. The fences allowed small rodents like mice and voles (mainly *Apodemus sylvaticus* and *Microtus arvalis*), abundant in the experimental area, to pass [Jessen & Eskelinen, personal observation,73]. In 2019, one plot originally intended for herbivore exclusion only was accidentally fertilized and one plot originally intended for herbivore exclusion and fertilization treatment was not fertilized. Both plots were treated as fertilized in the data analysis.

In addition to the fertilization and herbivore exclusion treatment we implemented a light addition treatment using modern LED lamps (C65, Valoya, Finland). For this purpose, the $1.4 \times 1.4$ m plots were divided into two $0.7 \times 1.4$ m subplots, one of which was randomly assigned to light addition treatment resulting in total 80 experimental subplots. Each lighted subplot was equipped with two 120 cm long and 3.5 cm wide high intensity LED lamps that were installed horizontally to the ground and parallel to each other. The light addition treatment started at the same time than fertilization treatment, i.e., at the end of May 2017. In the

beginning of the growing season (February–April) the lamps were installed about 10 cm above the smallest plants and uplifted as vegetation grew. We set the lamps to switch on two hours after sunrise and switch off two hours before sunset, and they automatically switched off when the temperature exceeded 28˚C to prevent overheating. In 2020, we switched on the lamps March 16. The light addition treatment added light throughout the active growing season and lamps were only removed during the grazing to prevent the sheep from damaging the lamps. Lamps in the fenced plots were switched off during the grazing. At the end of each growing season (October-November), the lamps were removed to prevent frost damage. Using LED lamps that have a wavelength spectrum corresponding to natural sunlight we mimicked increased sunlight availability in a vegetation gap in a natural grassland, and were therefore able to test the role of light limitation.

The sheep grazing on the pastures where our experiment is located are owned by a private shepherd and do graze the pastures freely. We are not taking any kind of measurements from the sheep nor do we or our experimental set up manipulate the sheep. Since no animal testing is involved in our experiment there was no need for approval by Institutional Animal Care and Use Committee (IACUC) or other relevant ethics board.

## Planting of *Silene*

In August 2019 we added seeds of *Silene* (local seeds from Saale-Saaten Stolle, Germany) in cultivation trays in a glasshouse. When the plants had grown for four months they were translocated into a cold house to allow them to adapt to outside conditions. In November 2019, when the plants had grown for five months, we planted one similar-sized individual into each experimental subplot. As *Silene* is a dioicous species and its genders have been found to differ in size [74], we allowed the transplants to grow relatively large before planting (i.e., they already had adult-sized large leaves and a flowering stalk). We then selected individuals that had similar-sized leaves and were equal in height, and randomly allotted these to the treatments. We are therefore convinced that gender differences should not affect our results and, if something, they could only weaken detecting significant responses to the treatments. In lighted subplots we made sure the transplant was planted directly under the LED lamp. After transplanting we watered the plants. By planting five-month-old relatively large individuals we also reduced the risk of vole/mice herbivory as smaller/younger plants are often eaten by voles and mice [75,76]. During winter, we further protected the transplants with round cylinders of green mesh (Netzhülle Bestklima, best4forst, Germany) with a height of 18 cm, a diameter of 12 cm and a mesh size of 3 × 3 mm. The mesh was attached to the ground to prevent voles/mice going in below the mesh. The protective mesh was removed in April 2020 when the growing season and plant growth started.

## Performance and trait sampling

To determine growth of *Silene* transplants, we measured vegetative height (cm) from the ground to the highest point of each plant. Height was measured after the plants have been grazed, but had the chance to regrow and compensate since harvesting took place two months after the last grazing event. Measuring height of grazed and ungrazed individuals two months after the last grazing allowed us to specifically test for the net effect of grazing, i.e., how much transplants compensated and were able to recover from grazing, under the combination of light addition, nutrient enrichment and climate change. We chose to measure trait variation in SLA and foliar C:N ratio because these have been shown to be associated with slow/fast resource uptake, responsiveness to light availability and attractiveness to herbivores [28,46,64,67]. For data on SLA ($mm^{-2}$ mg dry mass) we collected a maximum of three healthy

looking and well-developed leaves on each *Silene* transplant. SLA was measured directly after harvesting following the standard protocol [77,78] and averaged per transplant for the statistical analysis. We only sampled leaves that did not show any grazing or other damage. Since most of the leaves on transplants outside the fences were completely eaten, we could not measure foliar traits in grazed plots. Foliar traits were therefore only sampled from transplants inside the fences (i.e., from 40 individuals) and we could not test for a main nor interactive effect of grazing on the foliar traits. For the foliar C:N analysis we pooled the leaf material of each transplant and the chemical analysis was run with a VarioEL CNHS analyzer (Elemental Analyse Systeme GmbH, Hanau, Germany). After measurements of height measurements and collection of leaves, we harvested the aboveground biomass of *Silene*. Overall, 78 out of 80 individuals were still present at the point of harvest (i.e., after one year). The harvested plants were oven dried at 60˚C for 48 hours and then weighed to obtain data on aboveground biomass allocation as a surrogate for plant performance [7,11]. As with height measurements, we recorded biomass in grazed and ungrazed plots to assess how the net effects of grazing on *Silene* performance depended on nutrient and light addition and future climate. Contrary to *Silene* leaves, which we could not sample outside the fences, we could still obtain biomass data from *Silene* outside the fences. Both the trait sampling and the harvesting took place from the beginning to the middle of August 2020.

## Statistical analysis

To assess the net effect of grazing under nutrient and light addition, future climate, and their interactions on *Silene* growth and performance, we applied linear mixed effects models with height and biomass as response variables, both in their own models, and fencing, fertilization, light addition, climate treatments and their interactions as explanatory variables (fixed factors). Plot was nested within block as a random factor to account for the split-plot experimental design. To examine how nutrients, light addition, future climate and their interactions affected foliar traits of *Silene*, we applied similar linear mixed effect models, where SLA and foliar C:N were response variables, both in their own models, and fertilization, light addition, climate treatments and their interactions as explanatory variables (fixed factors), using similar random variable as in models above. Overall, we ran four different models. All models were restricted to three-way interactions to prevent the models from overfitting. The significance of the single factors and their interactions was tested with F-test. We used model diagnostic plots to check the homogeneity of variances and the normality of errors [79]. Height and biomass were log transformed to meet these assumptions.

We used the 'lme4' package [80] to run linear mixed effect models, the 'car' package for assessing the significance of the treatments effects [81], the 'tidyverse' package for data manipulation [82], the 'ggplot2' package [83] and the 'patchwork' package [84] for plotting in R statistical software [85, version 4.0.3].

## Results

### The responses of *Silene* performance to grazing, nutrient and light additions, and future climate

*Silene* grew tallest under future climate and when fertilized but not lighted (significant F x C x L interaction on height, Table 1, Fig 1A). This effect, as many other treatment effects, was especially pronounced inside fences, although there was no significant interaction with fencing (Table 1). Fertilization increased *Silene* height by 52% ($\sim$ 12 ± 4.4 cm) and aboveground biomass by 89% compared to unfertilized conditions (Table 1). *Silene* also attained greater

**Table 1. Main and interactive treatment effects on *Silene* performance.**

| Source of variation | Height[a] | | | Biomass[a] | | |
|---|---|---|---|---|---|---|
| | df | F | p | df | F | p |
| Exclosure | 1,24 | 103.76 | < .001 | 1,24 | 132.21 | < .001 |
| Fertilization | 1,25 | 5.61 | **0.026** | 1,25 | 5.51 | **0.027** |
| Climate | 1,8 | 1.83 | 0.214 | 1,8 | 2.28 | 0.171 |
| Light | 1,33 | 0.33 | 0.572 | 1,32 | 3.79 | *0.060* |
| E x F | 1,25 | 1.12 | 0.296 | 1,25 | < .1 | 0.948 |
| E x C | 1,24 | < .1 | 0.813 | 1,24 | 4.28 | **0.049** |
| E x L | 1,33 | < .1 | 0.966 | 1,32 | 1.00 | 0.326 |
| F x C | 1,25 | 2.43 | 0.132 | 1,25 | 1.50 | 0.232 |
| F x L | 1,33 | < .1 | 0.856 | 1,32 | 0.45 | 0.507 |
| C x L | 1,33 | < .1 | 0.824 | 1,32 | < .1 | 0.797 |
| E x F x C | 1,25 | 1.36 | 0.254 | 1,25 | < .1 | 0.795 |
| E x F x L | 1,33 | 2.34 | 0.136 | 1,32 | 0.77 | 0.388 |
| E x C x L | 1,33 | 1.91 | 0.176 | 1,32 | 2.52 | 0.122 |
| F x C x L | 1,33 | 6.67 | **0.014** | 1,32 | 0.56 | 0.461 |

Results of linear mixed effect models testing the effects of grazing (exclosure), fertilization, climate and light addition on *Silene* performance (height and total aboveground biomass). Significant results (p < .05) are printed in bold, marginally significant results (p < .1) are printed in italic.

[a] data are log transformed.

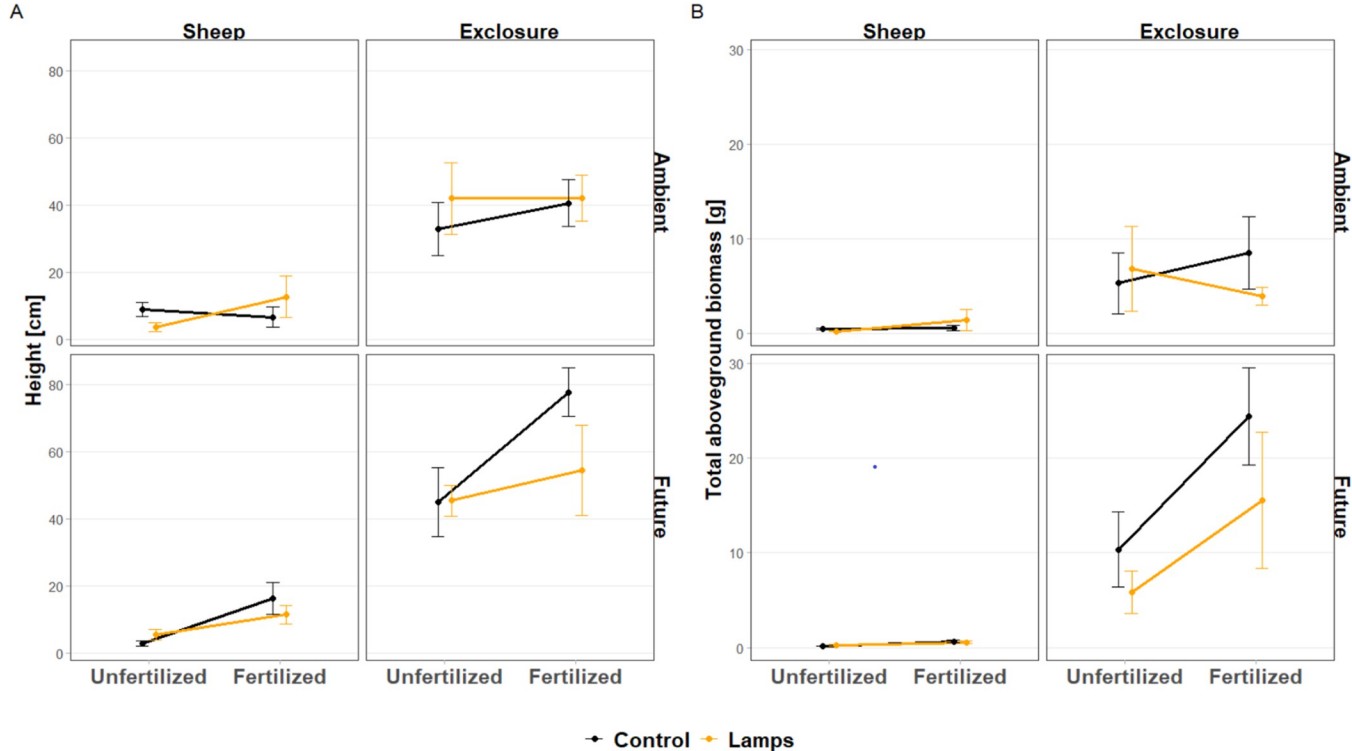

**Fig 1. Treatment effects on *Silene* performance.** Height (a, n = 80) and biomass (b, n = 78) responses of *Silene latifolia* to combinations of grazing, fertilization, ambient and future climate and light addition. The data are means ± SE. Note, that two transplants were not harvested for biomass determination because they were grazed to a few millimeters from the ground.

**Table 2. Main and interactive treatment effects on *Silene* foliar traits.**

|  | SLA_fenced | | | Foliar CN_fenced[a] | | |
|---|---|---|---|---|---|---|
|  | df | F | p | df | F | p |
| Fertilization | 1,8 | 0.23 | 0.644 | 1,11 | 5.84 | **0.034** |
| Climate | 1,7 | < .1 | 0.805 | 1,7 | 1.75 | 0.225 |
| Light | 1,11 | 7.01 | **0.023** | 1,16 | < .1 | 0.918 |
| F x C | 1,9 | 2.22 | 0.169 | 1,11 | 0.35 | 0.567 |
| F x L | 1,10 | < .1 | 0.861 | 1,16 | 5.96 | **0.027** |
| C x L | 1,11 | 5.56 | **0.038** | 1,16 | 1.35 | 0.262 |
| F x C x L | 1,11 | 1.71 | 0.218 | 1,16 | 0.90 | 0.357 |

Results of linear mixed effect models testing the effects of fertilization, climate and light addition on *Silene* leaf traits (SLA and foliar C:N). Significant results (p < .05) are printed in bold, marginally significant results (p < .1) are printed in italic.

[a] data are log transformed.

biomass in future climatic conditions compared to ambient conditions, however, only in the absence of grazing (significant E x C interaction, Table 1, Fig 1B). Grazing also showed a strong main effect and reduced plant height by 83% ($\sim$ 39 ± 3.6 cm) and total aboveground biomass by 95% compared to fenced plots (Table 1, Fig 1A and 1B).

## The responses of *Silene* foliar traits to nutrient and light additions, and future climate

Foliar leaf traits inside fences responded to resources additions. C:N ratio increased (i.e., the relative C content increased) with light addition in fertilized plots, while light addition had no effect on it in unfertilized plots (significant F × L interaction, Table 2, Fig 2A). Furthermore, SLA increased with light addition, but only under ambient climate (significant C × L interaction, Table 2). In addition, this effect was greater in fertilized plots although the interaction with fertilization was not significant (Fig 2B and Table 2). Light addition alone increased SLA

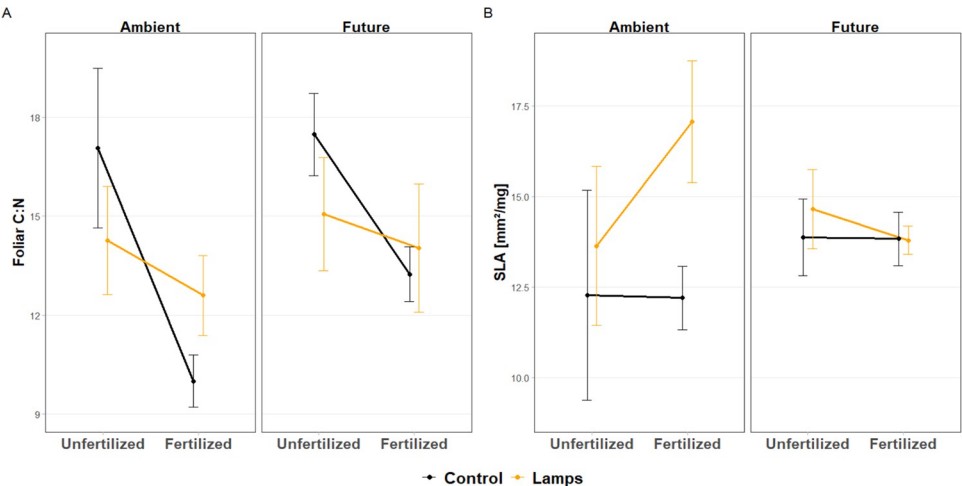

**Fig 2. Treatment effects on *Silene* foliar traits.** Foliar C:N (a, n = 36) and SLA (b, n = 32) responses of *Silene latifolia* to combinations of fertilization, ambient and future climate and light addition. The data are from inside the fences and represent means ± SE. Note, that there are less than 40 samples per trait, because some transplants did not have healthy and undamaged leaves.

by 12.8% (Table 2, S1 Fig) and fertilization alone decreased foliar C:N (i.e., increased relative foliar N content) by 21.9% (Table 2, Fig 2). Future climate had no main effect on *Silene* SLA and foliar C:N (Table 2).

## Discussion

Testing how single plant species respond to joint effects of nutrients and lights in combination with changes in grazing patterns and climate is important to better understand changes in plant community composition and diversity. We found that *Silene* grew taller under fertilized and future climatic conditions without additional light than when light was added. This finding demonstrates that competition for light is an important mechanism affecting plant investment into height under densely vegetated conditions, which has not been previously shown using direct manipulation of light in the field. Light was also an important modulator affecting SLA and C:N responses to climate and nutrients, highlighting that competition for light affects trait expression. In addition, future climate increased *Silene* biomass but this impact was conditional on the absence grazers, highlighting that individual species' responses to climate change can strongly depend on grazing. In general, strong responses to grazing highlight the important role of grazing animals for performance and traits of highly palatable forbs in grassland systems, especially under future climatic conditions.

We found that *Silene* transplants grew tallest under future climate when fertilized but not lighted. These effects were clearest in unfenced plots, as transplants were almost completely eaten when exposed to grazing. In these plots, general vegetation biomass was high and light levels low (S2A and S2B Fig, Eskelinen et al., submitted). This suggests that *Silene* invested in height in densely vegetated and shaded conditions (S3A and S3B Fig) as tall stature can increase the plant's ability to compete for light [13,86]. Because our experiment directly adds light into nutrient-enriched, fenced and climate-manipulated grassland communities, where light levels are low, we suggest that tall stature is a direct response to increased competition for light (S4 Fig). We also found that the increase in transplant height in fertilized and unlighted plots was limited to future climatic conditions. Increased spring precipitation and warmer temperatures in the future climate treatment may have promoted transplant growth during the early growing season and life stage [87,88] due to more favorable growing conditions than under ambient climate. Our finding suggests that changes in seasonal precipitation and temperature patterns can be a decisive factor for species' ability to grow tall and compete for light under altered nutrient and grazing conditions [56,89].

We also found that the addition of light increased SLA especially in fenced plots. This effect also depended on the climate treatment such that the positive response of SLA to light addition was prevalent in ambient climate only. Future climate with lower summer precipitation and higher temperature may have caused the transplants to invest less on expanding leaves during the summer months, resulting in lower SLA due to water scarcity [52]. Nevertheless, the observed increase in SLA in response to light addition seems counterintuitive, as plants are expected to increase SLA in response to light limitation [86]. However, also other factors than direct light intensity could have driven SLA responses which is supported by the fact that we did not find a significant relationship between light intensity and SLA (S5A Fig). Instead, SLA was positively related to percentage total vegetation cover (S5B Fig), suggesting that dense vegetation with high competition for space can be a decisive factor for foliar SLA manifestation. Competition for water, nutrients or space due to increased abundance of species with low SLA and leaf water content (LWC) (Eskelinen et al. submitted) may also contribute to explain the tendency of *Silene* total biomass to decrease in lighted plots. In addition, plants growing tall in dense vegetation may also grow thinner and thus produce less biomass [90]. Furthermore, also

foliar C:N responded to light addition and this effect was contingent on fertilization such that the positive effect of light addition on foliar relative carbon content was prevalent only in fertilized but not in unfertilized plots in the absence of grazers. This shows that high levels of nutrients and light can jointly promote foliar carbon investment. In a meta-analysis, Poorter, Niinemets [20] found that foliar carbon content increased in response to increasing light intensity and suggested that this was due to an increase of structural components such as lignin which are rich in carbon.

In addition, we found that future climate had a positive effect on the biomass of *Silene*; however, this positive effect was only evident in ungrazed plots as transplants were partly or completely eaten when not protected by fencing. Finding that herbivores can modulate plant responses to climate change is in line with previous studies in tundra which identified grazing as a key factor for mitigating climate warming effects on plant growth [11,91]. In particular, grazers have been found to limit the establishment of warm-adapted species [11,92], prevent upward shifts of treeline [93] and buffer diversity loss under warmer climate [94–96]. Overall, these findings provide evidence that the effect of grazing can have stronger implications on individual plant and plant community performance than climate change. To date, most of these studies have been from high latitude or high elevation systems. Our results extend the important role of herbivores as climate response modulators to temperate grasslands. This is especially important as natural grazing systems in temperate grasslands (e.g. in Central Germany) are declining, while climate change effects are increasing [97].

Grazing had the strongest effect on *Silene* height and biomass in our study. This finding is in line with previous research showing that grazers have strong short- and long-term effects on individual species traits and performance [8,11,26,56,98]. Reduced height and biomass are a direct consequence of plants being eaten. Addition of light and nutrients did not help plants to recover from grazing, suggesting that highly palatable plants to grazers, such as *Silene* [59,60], suffer from grazing regardless of resource supply and may not be able to compensate for the lost biomass at least during the same growing season. In our study, plants were harvested two months after the last grazing, which should have given some time to compensate. As a result, heavily grazed systems may lose species that are attractive to herbivores, especially if they do not enter the reproductive phase [99], as found in our study with only one flowering *Silene* individual in grazed plots.

In addition, we found that *Silene* individuals increased in height and biomass and their foliar C:N decreased in response to fertilization. This aligns with other studies investigating the effects of fertilization on individual species performance and traits [14,15]. In contrast to foliar C:N, SLA was unresponsive to fertilization. This is in line with Siefert and Ritchie [100] who found that the responsiveness of SLA to nutrient addition decreases with plant height and argue that tall-statured species are less adapted to shady conditions and therefore can be less plastic in their leaf traits [101]. This can explain why the SLA of *Silene*, which is a rather tall species, did not increase in fertilized plots.

In conclusion, our study highlights the role of light limitation on growth (i.e., height) allocation of individual plants in highly competitive (i.e., densely vegetated) environments. Our results also underscore that light availability interacts with climate and nutrients to affect plant foliar traits. Although we only studied one species, and not the overall community, *Silene* is representative of a common and palatable grassland forb, and may therefore provide insights in the growth and performance of similar grassland forbs. Our study also provides evidence that future climate is likely to boost plant growth, but only if plants are not exposed to grazing and therefore emphasizes the important role of grazers as modulators of plant performance under future climate. This knowledge is particularly important in an era of rapidly advancing climate change and concurrent decline in natural grazing systems, not only at high latitudes

but also at temperate latitudes [97], and ideally can support grassland management that incorporates grazing measures.

## Supporting information

**S1 Fig. Light addition effect on *Silene* SLA.** SLA under control and lighted conditions. The data are means ± SE from the leaves collected from inside the fences.
(TIF)

**S2 Fig. Treatment effects on vegetation and litter cover.** (a) Percent total vegetation cover and (b) percent litter cover in grazed (sheep) and ungrazed (exclosure), unfertilized and fertilized plots under ambient and future climate conditions. Data are means ± SE. Total vegetation cover and litter cover were assessed by visual estimation on the experimental plots in June 2020.
(TIF)

**S3 Fig. Regression of *Silene* height and vegetation and litter cover.** Regression of the visually estimated (a) total vegetation cover and *Silene* height and (b) litter cover and *Silene* height. The line represents a regression line with a 95% CI. The regressions are significant (a) $F_{1,74} = 7.58$, $P = 0.007$ (b) $F_{1,56} = 21.74$, $P < 0.001$. Note that the regression equations are based on log-transformed height values, while the y-axis shows not transformed values. Rm (marginal) refers to the amount of variance explained by the fixed effects only, Rc (conditional) refers to the amount of variance explained including the random effects (i.e. plot nested in block).
(TIF)

**S4 Fig. Regression of *Silene* height and light intensity.** Regression of the light intensity measured approximately 7–10 cm under the lamps and 15–20 cm above ground level and *Silene* height. The line represents a regression line with a 95% CI. The regression is significant $F_{1,67} = 5.30$, $P = 0.024$. Note that the regression equation is based on log-transformed height values, while the y-axis shows not transformed values. Rm (marginal) refers to the amount of variance explained by the fixed effects only, Rc (conditional) refers to the amount of variance explained including the random effects (i.e. plot nested in block).
(TIF)

**S5 Fig. Regression of *Silene* SLA and light intensity and litter cover.** Regression of (a) the light intensity measured approximately 7–10 cm under the lamps and 15–20 cm above ground level and *Silene* SLA and (b) the visually estimated percentage litter cover and *Silene* SLA. The lines represent regression lines with a 95% CI. The dashed regression line is not significant ($F_{1,29} = 0.38$, $P = 0.544$), the solid regression line is significant ($F_{1,44} = 13.67$, $P < 0.001$). Rm (marginal) refers to the amount of variance explained by the fixed effects only, Rc (conditional) refers to the amount of variance explained including the random effects (i.e. plot nested in block).
(TIF)

## Acknowledgments

We appreciate the Helmholtz Association, the Federal Ministry of Education and Research, the State Ministry of Science and Economy of Saxony-Anhalt and the State Ministry for Higher Education, Research and the Arts Saxony to fund the Global Change Experimental Facility (GCEF) project. We thank the staff of the Bad Lauchstädt Experimental Research Station (especially Ines Merbach and Konrad Kirsch) and Martin Schädler for their work in maintaining the plots and infrastructures of the Global Change Experimental Facility (GCEF), and

Harald Auge, François Buscot, Stefan Klotz, Thomas Reitz and Martin Schädler for their role in setting up the GCEF. We are grateful to Susanne Dunker for discussions when planning the experiment, to Peter Portius and the UFZ workshop staff for planning and manufacturing the lamp mounts, to Anna Leyendecker and Paul Kühn for supportive help in the field and to Daya Södje and Petra Hoffmann for valuable help in the lab.

## Author Contributions

**Conceptualization:** Anu Eskelinen.

**Data curation:** Maria-Theresa Jessen.

**Formal analysis:** Maria-Theresa Jessen.

**Funding acquisition:** Anu Eskelinen.

**Supervision:** Harald Auge, W. Stan Harpole, Yann Hautier, Anu Eskelinen.

**Visualization:** Maria-Theresa Jessen.

**Writing – original draft:** Maria-Theresa Jessen.

**Writing – review & editing:** Maria-Theresa Jessen, Harald Auge, W. Stan Harpole, Yann Hautier, Anu Eskelinen.

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
