## [Decision Letter · Decision Letter 0]

21 Apr 2022

PONE-D-22-06013Grazing and light modify Silene latifolia responses to nutrients and future climatePLOS ONE

Dear Dr. Jessen,

Thank you for submitting your manuscript to PLOS ONE. After careful consideration, we feel that it has merit but does not fully meet PLOS ONE’s publication criteria as it currently stands. Therefore, we invite you to submit a revised version of the manuscript that addresses the points raised during the review process.

We look forward to receiving your revised manuscript.

Kind regards,

Mai-He Li, Ph.D.

Academic Editor

PLOS ONE

Journal Requirements:

(This work was supported by the Flexpool program of the German Centre for Integrative Biodiversity Research (iDiv) – Halle, Jena, Leipzig [grant no. 34600565-11] and by a research grant from the Finnish Academy to A.E. [project 29719].

https://www.idiv.de/en/research/flexpool.html

https://www.aka.fi/en/

The funders had no role in study design, data collection and analysis, decision to publish, or preparation of the manuscript.)

(This study was funded by the iDiv Flexpool program (grant no. 34600565-11) and a Finnish Academy research grant (project 29719) to A.E. We appreciate the Helmholtz Association, the Federal Ministry of Education and Research, the State Ministry of Science and Economy of Saxony-Anhalt and the State Ministry for Higher Education, Research and the Arts Saxony to fund the Global Change Experimental Facility (GCEF) project. We thank the staff of the Bad Lauchstädt Experimental Research Station (especially Ines Merbach and Konrad Kirsch) and Martin Schädler for their work in maintaining the plots and infrastructures of the Global Change Experimental Facility (GCEF), and Harald Auge, François Buscot, Stefan Klotz, Thomas Reitz and Martin Schädler for their role in setting up the GCEF. We are grateful to Susanne Dunker for discussions when planning the experiment, to Peter Portius and the UFZ workshop staff for planning and manufacturing the lamp mounts, to Anna Leyendecker and Paul Kühn for supportive help in the field and to Daya Södje and Petra Hoffmann for valuable help in the lab.)

(This work was supported by the Flexpool program of the German Centre for Integrative Biodiversity Research (iDiv) – Halle, Jena, Leipzig [grant no. 34600565-11] and by a research grant from the Finnish Academy to A.E. [project 29719].

https://www.idiv.de/en/research/flexpool.html

https://www.aka.fi/en/

The funders had no role in study design, data collection and analysis, decision to publish, or preparation of the manuscript.)

Additional Editor Comments:

Dear authors,

Based on comments made by the reviewers and my own reading. my decision is Major Revision, and thus, please carefully consider comments made the reviewers, especially by Reviewer 1, and revise your paper.

I look forwarding to reading your revised version.

Kind regards,

Reviewers' comments:

Reviewer's Responses to Questions

**Comments to the Author**

1. Is the manuscript technically sound, and do the data support the conclusions?

Reviewer #1: No

Reviewer #2: Yes

2. Has the statistical analysis been performed appropriately and rigorously? 

Reviewer #1: No

Reviewer #2: Yes

3. Have the authors made all data underlying the findings in their manuscript fully available?

Reviewer #1: Yes

Reviewer #2: Yes

4. Is the manuscript presented in an intelligible fashion and written in standard English?

Reviewer #1: Yes

Reviewer #2: Yes

5. Review Comments to the Author

Reviewer #1: Overall, I find the experiment to be relatively novel and the research topic important. The experiment is a 4-factor global change experiment (climate by grazing exclusion by fertilization by light addition experiment), monitoring both species performance and "trait manifestation.” The 4-factor component of this experiment is interesting and has the potential to advance this field. I also found the experimental design component of this manuscript to be well described and explained, making it very clear to the reader.

I have a few major concerns. First, the results showed that there was a decrease in biomass when there was an addition of light (Figure 1:B). This does not match with the design section of this manuscript that discussed “light limitation.” If light is limited, I would expect there to be an increase in biomass when light was added. The decline in biomass with light addition was not clearly addressed throughout the rest of the manuscript. Based on your findings there is no significant evidence that light is limiting for this species, and it appears that the addition of light could actually be negative for this species. The decrease in biomass with light addition could be due to interactions with other species that respond more positively to light addition. You might consider exploring this aspect of your findings, ensuring that your results match the statements in the Discussion.

My second concern is with the interpretation of the grazing results. Grazing reduced biomass under all conditions, so that the responses to fertilization, light, and climate manipulation were not evident under grazed conditions. However, these responses were measured for only a single species, and in manipulated plots that represented a very small proportion of the grazed fields. Hence, if Silene is a palatable and preferred species, it might get eaten regardless of conditions under this grazing regime (as you stated in the Discussion). In contrast, since future conditions will reflect whole community responses, grazing patterns (and hence plant responses) could be very different. This is especially true if herbivore populations are responding dynamically to forage quantity and quality (vs being imposed by humans). My interpretation of these data is that the heavy level of the grazing treatment means that the data from the grazed fields are really very different from the data from the ungrazed plots. Hence, it's likely that the statistical analysis for biomass and height should be split into the grazed and ungrazed plots, the way you did for the traits (like Table 2).

Minor:

Line 58: SLA isn't defined yet. Would define what that means before use of the acronym.

Line 64-65: Add a citation to support your claim on why grassland systems are different.

Line 69: missing a period at the end of the paragraph

Line105: Again, usage of the acronym NPK before it is defined. This could be confusing for the reader.

Lines 112-118: At the end of the Introduction, hypotheses would be more compelling than questions. For instance, were you using the light addition treatment to test light limitation as a mechanism to explain negative responses to N addition in past experiments? Is there a way to frame the interactions among factors to get at mechanisms?

Line 132-133: "Rodents such as the European hare (Lepus europaeus) can enter the experimental area." Rabbits and hares are not rodents. You could substitute "Small mammalian herbivores.." for "Rodents.."

Line 205: The sentence would be more grammatically correct if "by" were replaced with "with"

Line 208: …when "the" growing season…

Line 235-236: "All models were restricted to three-way interactions to prevent the models from overfitting." But in the Results I see 4 way interactions in Table 1, summarizing responses for biomass and height. In contrast, in Table 2 for SLA and C:N, the data are analyzed separately for enclosed vs grazed plots. There should be one analysis procedure, so that the structures of Table 1 and Table 2 match. The approach should be clearly justified in the Methods.

Lines 326-327: "This shows that relaxation of nutrient and light limitations can jointly promote foliar carbon investment." Since light addition reduced Silene biomass, there isn't evidence for a relaxation of light limitation. This section and elsewhere in the Discussion needs to be revised to reflect the light results.

Line 332-333: "…transplants were partly completely eaten when not protected by fencing." This sentence might be missing "or."

Line 337-338: note that there were added citations here that didn't get formatted

Figure 1: Figure is blurry

Figure 2: Figure is blurry

S1: Confused as to why this was included. This seems to overlap with and not differ from Figure 2

Reviewer #2: This paper study plant responses to interactions among climate, nutrients, grazing and light. This research is helpful for evaluating ecosystem species competition, community composition and biodiversity.

The manuscript for this paper is technically sound, and the data support the conclusions. Appropriate and rigorous statistical analysis was performed in the manuscript of this paper. For example, the manuscript used a linear mixed-effects model for data analysis, nesting plots within blocks, and taking into account autocorrelations. All data available in the manuscript. The manuscript is presented in an intelligible manner and written in standard English, with minor grammatical errors.

However, this article has some deficiencies as follows:

1. In the abstract part, the author describes the important content of the paper very clearly, but the description is too cumbersome. It is recommended to refine the abstract part. For example, there is no need to emphasize “an abundant and widespread grassland forb in Europe” in the abstract.

2. Statistical analysis needs to be described in as much detail as possible so that other scholars can repeat it.

3. It is suggested that the description of the results section needs to be improved, and should be a perfect elaboration of the four scientific questions raised above.

In summary, my suggestion is to make a minor revision of this manuscript this manuscript cannot be accepted.

6. PLOS authors have the option to publish the peer review history of their article (what does this mean?). If published, this will include your full peer review and any attached files.

Reviewer #1: No

Reviewer #2: No

---

## [Author Response · Author response to Decision Letter 0]

20 Jun 2022

Response letter

All line numbers are referring to the revised manuscript with track changes.

Reviewer #1: 

Overall, I find the experiment to be relatively novel and the research topic important. The experiment is a 4-factor global change experiment (climate by grazing exclusion by fertilization by light addition experiment), monitoring both species performance and "trait manifestation.” The 4-factor component of this experiment is interesting and has the potential to advance this field. I also found the experimental design component of this manuscript to be well described and explained, making it very clear to the reader.

Our response: We thank the reviewer for appreciating our study in general and are grateful for a thorough review and suggestions that have led to an improved version of our manuscript. 

(1) I have a few major concerns. First, the results showed that there was a decrease in biomass when there was an addition of light (Figure 1:B). This does not match with the design section of this manuscript that discussed “light limitation.” If light is limited, I would expect there to be an increase in biomass when light was added. The decline in biomass with light addition was not clearly addressed throughout the rest of the manuscript. Based on your findings there is no significant evidence that light is limiting for this species, and it appears that the addition of light could actually be negative for this species. The decrease in biomass with light addition could be due to interactions with other species that respond more positively to light addition. You might consider exploring this aspect of your findings, ensuring that your results match the statements in the Discussion.

Our response: Thank you for this comment. We agree that our results do show a tendency of decreased biomass with light addition, however, this result is only marginally significant and was therefore not discussed extensively. Furthermore, we agree that light addition could have benefitted other species, particularly species with low SLA and water content, which we have shown in a recently submitted manuscript (Eskelinen et al. submitted). This could lead to higher competition for resources and result in Silene individuals with smaller biomass. We have added a sentence to the discussion (please see lines 345 – 348). However, there was no direct relationship between Silene biomass and light availability (F1,67 = 1,54, P = 0.219) and we should be cautious with interpreting any light effects on Silene biomass. 

Silene increased in height in unlighted, fertilized and future climate plots, where light limitation is likely to be highest (due to high total biomass). In such conditions, plants grow taller to avoid light limitation. This is confirmed by a regression analysis showing a significant negative relationship between Silene height and light intensity. We now added a figure showing a negative correlation between Silene height and light intensity, reinforcing our interpretation that lower light levels lead to greater investment on height (Fig. S4 in Supporting Information). It is possible that when growing taller in dense vegetation, plants also grow thinner and might have lower biomass. This could also explain the marginally significant negative effect of light on Silene biomass. We also added a sentence about this to the Discussion (see lines 348-349). 

(2) My second concern is with the interpretation of the grazing results. Grazing reduced biomass under all conditions, so that the responses to fertilization, light, and climate manipulation were not evident under grazed conditions. However, these responses were measured for only a single species, and in manipulated plots that represented a very small proportion of the grazed fields. Hence, if Silene is a palatable and preferred species, it might get eaten regardless of conditions under this grazing regime (as you stated in the Discussion). In contrast, since future conditions will reflect whole community responses, grazing patterns (and hence plant responses) could be very different. This is especially true if herbivore populations are responding dynamically to forage quantity and quality (vs being imposed by humans). My interpretation of these data is that the heavy level of the grazing treatment means that the data from the grazed fields are really very different from the data from the ungrazed plots. Hence, it's likely that the statistical analysis for biomass and height should be split into the grazed and ungrazed plots, the way you did for the traits (like Table 2).

Our response: Thank you for raising this important concern. The rationale for including the effects of grazing in the analysis of Silene performance (i.e., height and biomass) is to evaluate the net effect (i.e., after two months since the last grazing) of short-term intensive grazing under changing environmental conditions such as future climate or nutrient enrichment, as this will allow conclusions to be drawn about possible recovery or compensatory growth of this species. We have emphasized this also in the Methods (please see lines 223-226). It was one of our main interests to examine how grazing interacts with nutrients, light and future climate, and we would not be able to address our main questions using separate data sets for fenced and unfenced plots. Some of our main findings also rely on interactions between grazing and other treatments (e.g., how grazing mitigates the positive effects of future climate on Silene performance). 

Further, since the experimental design is full-factorial (fencing, nutrient addition, light addition, future climate), it would be against fundamental principles to split the data for separate analyses. Splitting the data would also considerably reduce statistical power. 

For the leaf trait data, the situation is completely different, since there was no sufficient amount of leaves present to be collected from the grazed plots, so we simply missed observations completely. We only have observations from ungrazed plots and could not assess the impact of grazing. We have highlighted this more in the Methods section (please see lines 232-234).

We agree with the reviewer that responses of less palatable species and thus also overall community responses can deviate from our results and have therefore highlighted that our results present responses of Silene latifolia and similar species only (please see lines 38-39, 390-392). 

Our experiment is confined to short-term high intensity grazing which is suggested

as conservation tool to maintain open grasslands in Germany (LLUR, 2010) and should therefore inform about plant responses under a locally common grazing practice. We added information about this type of grazing to the Methods (please see lines 139– 140). In general, the sheep graze the entire pastures (16 x 24 m) in which our experimental plots (1.4 x 1.4 m) are embedded. These pastures also contain naturally occurring Silene and other highly palatable forbs (e.g., Daucus carota, Knautia arvensis, Centaurea jacea.) such that the grazing pressure is not exclusively concentrated on the plants in the experimental plots. 

Minor:

(3) Line 58: SLA isn't defined yet. Would define what that means before use of the acronym.

Our response: We have defined SLA already in the abstract (please see line 29).

(4) Line 64-65: Add a citation to support your claim on why grassland systems are different.

Our response: Thank you for this suggestion, we have added a reference (please see line 65).

(5) Line 69: missing a period at the end of the paragraph

Our response: Thank you for noting the error – added. 

(6) Line 105: Again, usage of the acronym NPK before it is defined. This could be confusing for the reader.

Our response: Thank you for noticing this, we have spelled out NPK (please see line 106).

(7) Lines 112-118: At the end of the Introduction, hypotheses would be more compelling than questions. For instance, were you using the light addition treatment to test light limitation as a mechanism to explain negative responses to N addition in past experiments? Is there a way to frame the interactions among factors to get at mechanisms?

Our response: Thank you for this suggestion. We have formulated our research questions more precisely and have added predictions and hope that this will allow the underlying relationships to be understood more clearly (please see lines 113-125).

(8) Line 132-133: "Rodents such as the European hare (Lepus europaeus) can enter the experimental area." Rabbits and hares are not rodents. You could substitute "Small mammalian herbivores.." for "Rodents.."

Our response: We apologize for the incorrect wording and have substituted “rodents” with “small mammalian herbivores” (please see lines 141-142).

(9) Line 205: The sentence would be more grammatically correct if "by" were replaced with "with"

Our response: We have replaced “by” with “with” (line 215).

(10) Line 208: …when "the" growing season…

Our response: We have added “the” (line 218).

(11) Line 235-236: "All models were restricted to three-way interactions to prevent the models from overfitting." But in the Results I see 4 way interactions in Table 1, summarizing responses for biomass and height. In contrast, in Table 2 for SLA and C:N, the data are analyzed separately for enclosed vs grazed plots. There should be one analysis procedure, so that the structures of Table 1 and Table 2 match. The approach should be clearly justified in the Methods.

Our response: We apologize for this inconsistency and have adjusted Table 1 to only include all two- and three-way interactions. As mentioned in our response to comment 2, it was not possible to use the full data set for the leaf traits, because there were no leaves left on Silene in the grazed plots. This information is mentioned in the Methods section (lines 232-235 and 241-245).

(12) Lines 326-327: "This shows that relaxation of nutrient and light limitations can jointly promote foliar carbon investment." Since light addition reduced Silene biomass, there isn't evidence for a relaxation of light limitation. This section and elsewhere in the Discussion needs to be revised to reflect the light results.

Our response: Thank you for this comment. We have now been more careful in phrasing in Discussion. For example, in line 352 we now say, “high levels of nutrients and light”, instead of “relaxation of nutrient and light limitation”. 

(13) Line 332-333: "…transplants were partly completely eaten when not protected by fencing." This sentence might be missing "or."

Our response: We have modified the sentence according to the reviewer’s suggestion (please see line 357).

(14) Line 337-338: note that there were added citations here that didn't get formatted

Our response: Thank you for noticing, we have reformatted the references.

(15) Figure 1: Figure is blurry

Figure 2: Figure is blurry

Our response: We have changed the resolution of the figures and hope that their quality is better now.

(16) S1: Confused as to why this was included. This seems to overlap with and not differ from Figure 2

Our response: Figure 2 shows main and interactive effects of nutrient and light addition, and climate treatment on Silene SLA. Figure S1 plots SLA only in lighted and unlighted plots. Thus, Figure S1 does overlap with Figure 2 but was added to the Supporting Information to visualize the main effect of light addition more clearly and to specifically address and clarify this result which is the new aspect of our study. However, we can delete Figure S1 if the main effect of light is already evident for the reader in Figure 2.

References:

Landesamt für Landwirtschaft, Umwelt und ländliche Räume des Landes Schleswig-Holstein (LLUR). Beweidung von Offen- und Halboffenbiotopen. Eine adäquate Pﬂegemethode unter besonderer Berücksichtigung der FFH-Lebensraumtypen und Arten. Schriftenreihe des Landesamtes für Landwirtschaft, Umwelt und ländliche Räume des Landes Schleswig-Holstein, 18, 1–30 (2010). 

Reviewer #2

This paper study plant responses to interactions among climate, nutrients, grazing and light. This research is helpful for evaluating ecosystem species competition, community composition and biodiversity. 

The manuscript for this paper is technically sound, and the data support the conclusions. Appropriate and rigorous statistical analysis was performed in the manuscript of this paper. For example, the manuscript used a linear mixed-effects model for data analysis, nesting plots within blocks, and taking into account autocorrelations. All data available in the manuscript. The manuscript is presented in an intelligible manner and written in standard English, with minor grammatical errors.

Our response: We thank the reviewer for appreciating our study design, analyses and interpretation of the results.

However, this article has some deficiencies as follows:

(17) In the abstract part, the author describes the important content of the paper very clearly, but the description is too cumbersome. It is recommended to refine the abstract part. For example, there is no need to emphasize “an abundant and widespread grassland forb in Europe” in the abstract.

Our response: Thank you for this remark. We simplified the abstract as suggested (please see lines 27, 31-32, 35).

(18) Statistical analysis needs to be described in as much detail as possible so that other scholars can repeat it.

Our response: We have added more details to the statistical analysis section (please see lines 248-261).

(19) It is suggested that the description of the results section needs to be improved, and should be a perfect elaboration of the four scientific questions raised above.

In summary, my suggestion is to make a minor revision of this manuscript this manuscript cannot be accepted.

Our response: We appreciate this comment and apologize for the previous inconsistency of questions and result section. We have now reformulated the questions and added predictions based on a comment made by reviewer 1. The reworded questions reflect the structure of the results section much better (please see lines 113-116 in the Introduction for the reformulated questions).

---

## [Decision Letter · Decision Letter 1]

15 Aug 2022

PONE-D-22-06013R1Grazing and light modify Silene latifolia responses to nutrients and future climatePLOS ONE

Dear Dr. Jessen,

Thank you for submitting your manuscript to PLOS ONE. After careful consideration, we feel that it has merit but does not fully meet PLOS ONE’s publication criteria as it currently stands. Therefore, we invite you to submit a revised version of the manuscript that addresses the points raised during the review process.

We look forward to receiving your revised manuscript.

Kind regards,

Mai-He Li, Ph.D.

Academic Editor

PLOS ONE

Journal Requirements:

Additional Editor Comments (if provided):

Dear authors,

because the experts who previously reviewed your manuscript rejected to read your revised manuscript, and thus I took time to find other experts to review your paper. Finally, I got the very positive comments and thus I am pleased to accept your paper provisionly. Before final publication, a minor revision, according to comments made by the reviewer, is still needed.

Kind regards,

Mai-He Li

Reviewers' comments:

Reviewer's Responses to Questions

**Comments to the Author**

1. If the authors have adequately addressed your comments raised in a previous round of review and you feel that this manuscript is now acceptable for publication, you may indicate that here to bypass the “Comments to the Author” section, enter your conflict of interest statement in the “Confidential to Editor” section, and submit your "Accept" recommendation.

Reviewer #3: (No Response)

2. Is the manuscript technically sound, and do the data support the conclusions?

Reviewer #3: Yes

3. Has the statistical analysis been performed appropriately and rigorously? 

Reviewer #3: Yes

4. Have the authors made all data underlying the findings in their manuscript fully available?

Reviewer #3: Yes

5. Is the manuscript presented in an intelligible fashion and written in standard English?

Reviewer #3: Yes

6. Review Comments to the Author

Reviewer #3: The authors have conducted an effective factorial design to assess a herb species’ response to select environmental changes using transplanted individuals in a test setting. The results are interesting, demonstrating some expected and some unexpected results. I have no overarching concerns.

The end of the Introduction begins to encroach on Methods (perhaps lines 104 to 113), it seems, citing the ways in which extra light is delivered, traits measured, and data collected. This may be revisited to reduce duplication with Methods.

Line 116. Remove the comma.

Lines 116-119. It is odd that hypothesis 1a and an “unless” in it, so to speak. It a little like “’A’ will be true, unless it isn’t.” Not important, but a little unusual. Hypothesis 1b is weak as well, citing both sides of a response, so to speak. I’d suggest editing these to be declarative, citing a specific expected response. Hypothesis 2a is an example of such a statement.

Line 130. Add an “E” to the longitude.

Line 350. “absence of grazers”

Regression equations and percent variation explained would be helpful for the later supplemental figures.

7. PLOS authors have the option to publish the peer review history of their article (what does this mean?). If published, this will include your full peer review and any attached files.

Reviewer #3: No

---

## [Author Response · Author response to Decision Letter 1]

13 Oct 2022

Response letter to PONE-D-22-06013R1

Reviewer #3: The authors have conducted an effective factorial design to assess a herb species’ response to select environmental changes using transplanted individuals in a test setting. The results are interesting, demonstrating some expected and some unexpected results. I have no overarching concerns

We thank the Reviewer for appreciating our study and for a thoughtful and thorough review. We have done our best to improve the manuscript based on the Reviewer’s comments. Please see our detailed responses below. 

(1) The end of the Introduction begins to encroach on Methods (perhaps lines 104 to 113), it seems, citing the ways in which extra light is delivered, traits measured, and data collected. This may be revisited to reduce duplication with Methods.

Our response: Previous reviewers suggested adding this information to better understand the questions and predictions. As our experimental setup is rather complex, we agree with these reviewer suggestions and would like to keep the brief overview of the experimental treatments. Furthermore, we think it is important for the reader to understand our definition of transplant performance, before reading the questions. We have therefore decided to not change this part of the manuscript (please see lines 108 - 112) but are open for editorial advice in this matter.

(2) Line 116. Remove the comma.

Our response: Done.

(3) Lines 116-119. It is odd that hypothesis 1a and an “unless” in it, so to speak. It a little like “’A’ will be true, unless it isn’t.” Not important, but a little unusual. Hypothesis 1b is weak as well, citing both sides of a response, so to speak. I’d suggest editing these to be declarative, citing a specific expected response. Hypothesis 2a is an example of such a statement.

Our response: Thank you for this comment. We fully agree with the reviewer with regard to the prediction 1a and have now specified the expected response (please see lines 116-118). Prediction 1b specification is not straightforward because our climate treatment can lead to different outcomes depending on which season is the most influential for plant growth and survival (these scenarios are explained earlier in the Introduction, please see lines 85-91). We think it is therefore reasonable to integrate both possibilities in the prediction as we have done in prediction 2b.

(4) Line 130. Add an “E” to the longitude

Our response: Thank you for noting this: done.

(5) Line 350. “absence of grazers

Our response: We have added “absence of grazers”.

(6) Regression equations and percent variation explained would be helpful for the later supplemental figures.

Our response: Thank you for this suggestion. We have added the information to the figures in the Supporting Information.

---

## [Editor Report · Decision Letter 2]

14 Oct 2022

Grazing and light modify Silene latifolia responses to nutrients and future climate

PONE-D-22-06013R2

Dear Dr. Jessen,

We’re pleased to inform you that your manuscript has been judged scientifically suitable for publication and will be formally accepted for publication once it meets all outstanding technical requirements.

Kind regards,

Mai-He Li, Ph.D.

Academic Editor

PLOS ONE